

# WaterBench: A Large-scale Benchmark Dataset for Data-Driven Streamflow Forecasting

Ibrahim Demir[1,2], Zhongrun Xiang[1], Bekir Demiray[3], Muhammed Sit[3]

[1] Department of Civil and Environmental Engineering, University of Iowa, Iowa City, 52246 USA
[2] Department of Electrical and Computer Engineering, University of Iowa, Iowa City, 52246 USA
[3] Interdisciplinary Graduate Program in Informatics, University of Iowa, Iowa City, 52246 USA

*Correspondence to*: Zhongrun Xiang (zhongrun-xiang@uiowa.edu)

**Abstract.** This study proposes a comprehensive benchmark dataset for streamflow forecasting, WaterBench, that follows FAIR data principles that is prepared with a focus on convenience for utilizing in data-driven and machine learning studies,
and provides benchmark performance for state-of-art deep learning architectures on the dataset for comparative analysis. By aggregating the datasets of streamflow, precipitation, watershed area, slope, soil types, and evapotranspiration from federal agencies and state organizations (i.e., NASA, NOAA, USGS, and Iowa Flood Center), we provided the WaterBench for hourly streamflow forecast studies. This dataset has a high temporal and spatial resolution with rich metadata and relational information, which can be used for varieties of deep learning and machine learning research. We defined a sample streamflow
forecasting task for the next 120 hours and provided performance benchmarks on this task with sample linear regression and deep learning models, including Long Short-Term Memory (LSTM), Gated Recurrent Units (GRU), and S2S (Sequence-to-sequence). To some extent, WaterBench makes up for the lack of unified benchmarks in earth science research. We highly encourage researchers to use the WaterBench for deep learning research in hydrology.

## 1 Introduction

Deep learning, a set of artificial neural networks (ANN) based algorithms for supervised and unsupervised modeling, has been widely used and recognized as a powerful approach within many scientific disciplines for technological and predictive progress (Goodfellow et al., 2016). As conventional machine learning techniques deemed limited in learning the representations of high-dimensional datasets from their raw form, by providing universal approximator models (Cybenko, 1989; Hornik et al., 1989; Leshno et al., 1993), deep neural networks increased scientists' ability in modeling both linear and non-linear problems
without time-intensive data engineering processes by domain experts (LeCun et al., 2015). The power of deep learning in problem-solving has opened ways to advancements in many fields that machine learning has been a go-to solution for predictive modeling, such as image recognition and synthesis (Demiray et al., 2021), speech recognition, language modeling and time-series prediction.

Flooding is a significant concern for many areas in the world as it is on an upward trend due to climate change. The 1998
Bangladesh flood, the Iowa flood of 2008, and the 2013 North India floods show how catastrophic and both economically and psychologically devastating floods can be for populations in respective regions. In order to maximize the preparedness for



floods and minimize their effects after the disaster (Yildirim and Demir, 2021), weather and flood forecasting stands as a perennial research interest for hydrologists and data scientists. Flood forecasting (also known as streamflow prediction or runoff forecasting) is a modeling effort where the water height change of a stream over time is being modeled and forecasted

using previous data points for a location or nearby locations with similar characteristics. Although this effort is conventionally carried out with physically based models that require extensive computational (Agliamzanov et al., 2020) and data resources, it is critical for flood mitigation and decision support (Xu et al., 2020).

Being a time-series prediction task, in essence, flood forecasting takes advantage of the practicality and efficacy deep learning brings to predictive modeling. Both time-series adaptations of deep learning models intended for natural language processing,

and time-series focused deep neural network implementations make this possible by proposing methodologies that put the sequential nature of time-series datasets into good use. Recurrent neural network (RNN) architectures such as Long short-term memory (LSTM) networks (Hochreiter and Schmidhuber, 1997) and Gated recurrent unit (GRU) networks (Chung et al., 2014), and Attention based sequence-to-sequence networks (Vaswani et al., 2017) are pronounced starting point deep neural network architectures for most time-series forecasting tasks.

Supervised learning, whether it be deep or not, is the most common form of machine learning (LeCun et al., 2015), and supervised learning tasks, such as flood forecasting, need a dataset of previously recorded/labeled entries for the task. That dataset typically consists of X and y values where X values are the input that the model expects, and y values are the output values the model returns. A supervised learning model is trained using a loss function that measures the similarity or difference of the y values from the dataset (actual ys) and the outputs of the model (predicted ys). During a typical training process,

predicted ys get closer to the actual ys in time, hence the name training. As a quintessential part of any supervised learning task, training neural network models on established datasets is common among deep learning practitioners and researchers (Goodfellow et al., 2016). For most tasks that deep learning researchers tackle today, there are vast amounts of benchmark datasets available freely for research. While computer vision datasets such as Imagenet (Deng et al., 2009), Ms-celeb-1m (Guo et al., 2016), Adobe-240fps (Su et al., 2017), and Vimeo-90K (Xue et al., 2019) and similarly time-series datasets namely,

automobile parts demand dataset Parts (Seeger et al., 2016), electricity and traffic (Yu et al., 2016) have been widely used to test proposed neural network architectures, to the best of our knowledge. There are not many specific datasets that are published for geoscience studies (Ebert-Uphoff et al., 2017) and specifically flood and streamflow forecasting.

The number of studies in hydrology and water resources, and particularly in flood forecasting that employ deep learning, have been gaining interest in the last several years (Sit et al., 2020). Flood forecasting studies in the literature, due to the

aforementioned sequential nature, have been vastly employing RNNs and LSTMs. Kratzert et al. (2018) utilize LSTM networks for daily runoff prediction using meteorological datasets. Furthermore, Kratzert et al. (2019) apply a similar approach for ungauged US locations. Ni et al. (2020) combine Convolutional neural networks (CNNs) (LeCun, 1989) with LSTM networks and compare the results with a wavelet-based LSTM model. Similarly, Kabir et al. (2020) take advantage of wavelets and propose a wavelet-based ANN model for hourly stage measurements. Another study that uses CNNs (Wang et al., 2019)

utilizes satellite imagery to predict hourly stage height in real-time during typhoon season.





On a different approach, Bai et al. (2019) incorporate a stack autoencoder (SAE) with LSTM for daily streamflow measurements from data for a week. Xiang et al. (2020) predict the next 24-hours of hourly measurements by utilizing an encoder-decoder sequence-to-sequence neural network that also uses rainfall products. Xiang and Demir (2020), moreover, extend their study and develop a model that forecasts hourly measurements for the next five days using three days of historic

data. They also incorporate upstream sensors into their proposed network. Using the same dataset, Xiang et al., 2021, explore the generalization of sequence-to-sequence encoder-decoder networks in flood forecasting. Sit and Demir (2019) predict hourly sensor measurements for 24 hours using data from the upstream sensor network and historic stage height measurements. And finally Sit et al. 2021a, utilizes graph neural networks for streamflow forecasting for a small watershed in Iowa. Most of the studies mentioned here acquire several raw data products, whether in terms of rainfall measurements, physical features of

the studied area, or stage height/discharge measurements, from authorities and build their own dataset benefiting from their expertise in the area.

There are several datasets and benchmarks in other earth science studies, i.e., air quality forecast dataset, 3D cloud detection dataset, and LANL earthquake prediction dataset. One of the early user-friendly datasets in earth science is the Beijing PM2.5 Data. It was published in 2017, and it includes the hourly air quality PM2.5 data of the U.S. Embassy in Beijing and

meteorological data from Beijing Capital International Airport. After the dataset is released, researchers have developed different novel machine learning and deep learning models, including the support vector machines (Zhu et al., 2018, Liu et al., 2019), recurrent neural networks (Athira et al., 2018), attention-based LSTM (Li et al., 2019), interpretable deep learning (Guo et al., 2018),hybrid deep learning (Du et al., 2018),convolutional networks (Tao et al., 2019), and stacked LSTM (Sagheer and Kotb, 2019) on this specific dataset. This dataset solves the difficulty of data acquisition and does not require domain

knowledge from meteorology. Furthermore, these papers used the same dataset, and therefore, the results are comparable. Thus, scientists could focus more on modeling and improving on the basis of existing papers rather than collecting their own datasets. A benchmark in hydrology will no doubt enhance the application and development speed of deep learning studies in the water resources field.

Scientific advancement, intrinsically, is supposed to be cumulative, and in order to have better generalized deep learning-based

flood forecasting models, scientists need to build on top of what their fellow researchers have done. We believe that this could only be done by using the same set of testing mechanisms, and a common testing mechanism could only be achieved by using the same dataset when testing the capabilities of flood forecasting models. There are some studies in the literature of hydrology in limited numbers that construct their neural network architecture around the CAMELS dataset (Newman et al., 2014). CAMELS is a vast dataset that includes meteorological and observed streamflow data points for the United States, albeit not

in an easy-to-use and ideal format for deep learning research. It contains 671 catchments in the contiguous US that are minimally impacted by human activities. It includes the features such as the topography, climate, streamflow, land cover, soil, and geology in watershed scale, and the hydrometeorological time-series data ranges from 1980 to 2014 on a daily basis. The data is generated from different sources, including Daymet, NLDAS, and Maurer. CAMELS aggregated these datasets into the watershed level. The researchers also did the model simulation using physically-based models such as the NWS model,





and SNOW-17/SAC-SMA; however these modeling results are not shared as a benchmark. Even though there is a dataset that could be used for predictive deep learning rainfall-runoff modeling, there is still a lack of accessible datasets for benchmarking purposes (Masley et al., 2020). There remains a need for a dataset that is more convenient to use in deep learning research given that most of the deep learning researchers are not domain experts. The limited usage of CAMELS in the literature also predicates the challenges the CAMELS dataset presents for deep learning research.

Another dataset for flood forecasting is FlowDB (Godfried et al., 2020). Unlike CAMELS, there are not many studies that report their performance over FlowDB yet as the dataset is recently published. FlowDB is an hourly precipitation and river flow dataset that also includes a subset dataset for flash floods. The subset dataset includes injury costs and damage estimations for flash flood events. FlowDB gathers river flow data from USGS and precipitation data from many agencies, including USGS, NOAA, and ASOS. Additionally, the data FlowDB provides regarding flash floods uses NSSL Flash by NOAA.

This study proposes a flood forecasting dataset that follows FAIR data principles that is prepared with a focus on convenience for utilizing in data-driven and machine learning studies and provides benchmark performance for state-of-art deep learning architectures on the dataset for comparative analysis. WaterBench provides data from 125 catchments in the state of Iowa. The precipitation time-series data ranges from October 2011 to September 2018 along with catchment-based features such as the topography, soil type, and slopes. Even though the dataset was designed in a way to eliminate most of the preprocessing and

data engineering tasks out of the way for machine learning applications and research, it could be used in other studies with similar goals, such as physically based modeling with physical equations. Similarly, the dataset could be used by combining with other benchmark datasets such as IowaRain (Sit et al., 2021b) utilizing cloud-based rainfall products (Seo et al., 2019). WaterBench is different from CAMELS with a higher temporal resolution. In addition, it focuses on the state of Iowa, and many large catchments in WaterBench contain multiple USGS gauges, which helps to better represent the river structure, and

upstream-downstream relation in deep learning algorithms. The WaterBench is not selected based on human activities, which is a reaction to the real situation in Iowa. The rest of this paper is structured as follows; the dataset preparation phase and methodology employed in that phase are discussed in section 2. Section 3 gives a list of tasks that could be tackled using this dataset and presents the performance of several neural network implementations in flood forecasting tasks. In the last section, conclusions are discussed.

## 2 Methodology and Dataset

### 2.1. Study Area

The State of Iowa is located in the Midwest of the United States. It has abundant and diversified water resources with 71,655 miles of rivers and streams from border to border (Iowa DNR, 2004). In 2008, the Eastern Iowa was devastated with flooding which caused over $6 billion in property loss. The streamflow monitoring and forecasting are consequently critical for Iowa

for better water resources and disaster management. In addition, agricultural-based activities in Iowa have a low pavement rate with limited human influence, which makes it a suitable area for rainfall-runoff studies.

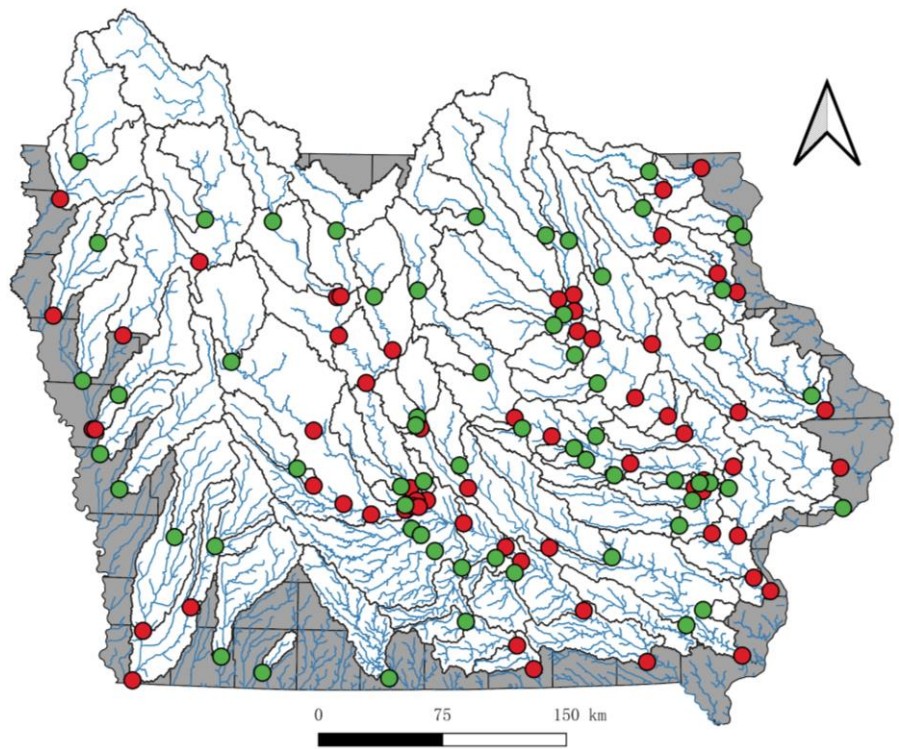

**Figure 1. The location of 125 USGS gauges in the State of Iowa with single (green dot) or multiple (red dot) stream gages.**

The United States Geological Survey (USGS) has over hundred streamflow gauges in the state of Iowa for monitoring the
streamflow rate in different streams. The measurements from the USGS are typically recorded at 15- to 60- minute intervals
in Iowa. Due to the site maintenance or shutdowns, coverage of the USGS streamflow gauges changes over the years. In this
dataset, we selected all USGS gauges in the State of Iowa with available data from October 1st, 2011 (the water year 2012) to
September 30th, 2018 (the water year 2018).

As shown in Figure 1, each USGS gauge is represented in green dots, and each dot monitors different catchments shown in
white background and black boundary. In large watersheds (red dot), multiple USGS gauges are located in the same stream,
and the watershed is split into multiple catchments. Thus, considering the connectivity of the streams, the relationship of these
gauges in one watershed can be represented as a tree structure. Green points represent the upstream watersheds with only 1
stream gauge located at the stream outlet. Red points represent the downstream watersheds with 1 stream gauge located at the
stream outlet, and 1 or more stream gauge located inside of the watershed.

**2.2. Dataset Features**

WaterBench includes detailed metadata and time-series features for each catchment. These datasets are available in .csv format
for each catchment. The details of the datasets with data source, type, resolution and units are shown in Table 1. The statistics
of the metadata, including the watershed size, concentration-time (the longest streamflow path in the catchment), slope, and





four soil types, are shown in Table 2 and Figure 2. The metadata, including the area, slope, travel time, and slope, are constant
for each catchment, and the streamflow, precipitation, and ET are in time series.

**Table 1. The details of datasets with data source, type, resolution and units**

| Datasets | Data Type | Sources | Resolution | | Unit |
|---|---|---|---|---|---|
| | | | Spatial | Temporal | |
| Area | GIS shapefile | IFC (Krajewski et al., 2017) | Station based | constant | km$^2$ |
| Slope | Hillslope data | | Hillslope based | constant | % |
| Travel time | Reach shapefile | | Station based | constant | hour |
| ET | Estimation from historical data | | State based | monthly | mm / month |
| Soil types | Soil data | NASA (Post et al., 2000) | 0.5-degree grid | constant | % |
| Streamflow Rate | USGS gage measurement | USGS | Station based | 15-60 mins | ft$^3$/s |
| Precipitation | Stage IV multi-sensor measurement | NOAA (Lin, 2011) | 4km grid | hourly | mm/hr |

**Table 2. The minimum, maximum, median, and standard deviation (SD) of the watershed area, concentration time, average slope,
and percentage of soil types including loam, silt, sandy clay loam, and silty clay loam among 125 USGS gauges in the State of Iowa.**

| | Area (km$^2$) | Concentration Time (hr) | Slope | Loam | Silt | Sandy clay loam | Silty clay loam |
|---|---|---|---|---|---|---|---|
| **Min** | 6 | 2 | 0.38% | 0% | 0% | 0% | 0% |
| **Max** | 36,453 | 315 | 4.32% | 98% | 100% | 84% | 93% |
| **Mean** | 5,405 | 77 | 1.97% | 33% | 31% | 18% | 18% |
| **Median** | 1,918 | 53 | 1.80% | 33% | 21% | 4% | 7% |
| **SD** | 8,320 | 68 | 0.80% | 28% | 30% | 24% | 23% |





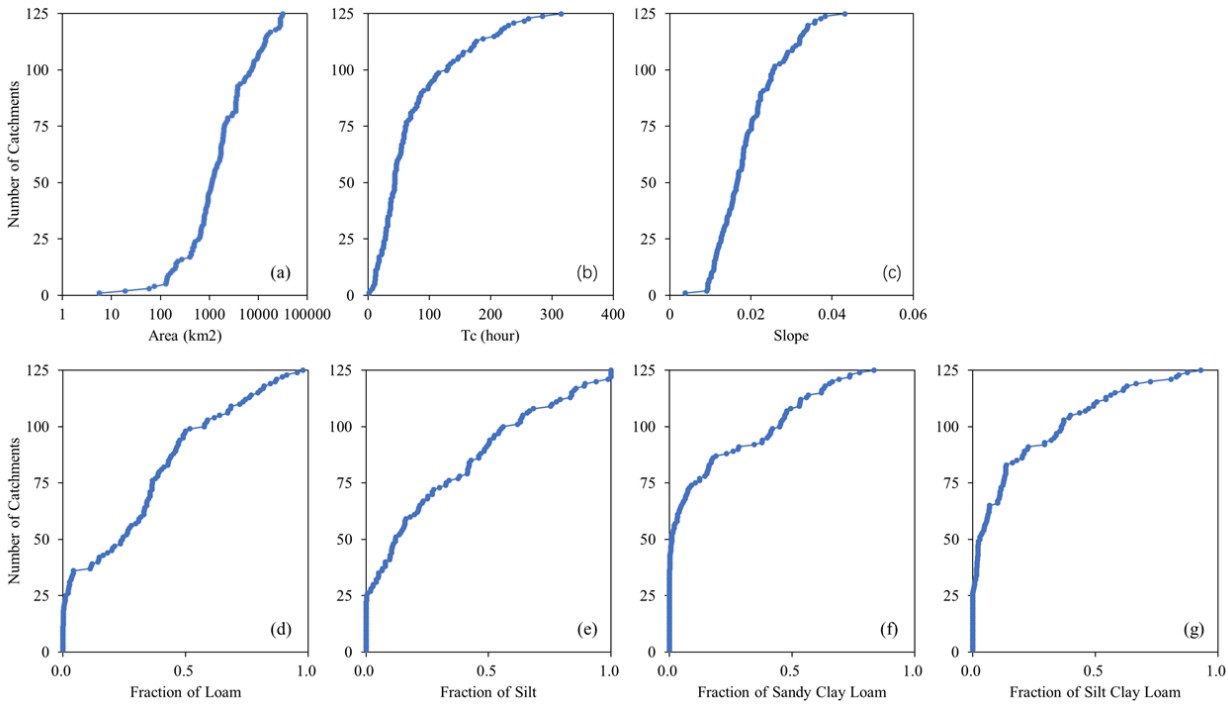

**Figure 2. Cumulative distribution of the catchment area (a), concentration time (b), average slope (c), and percentage of soil types including loam (d), silt I, sandy clay loam (f), and silty clay loam (g) for 125 USGS gauges in the State of Iowa.**

**Table 3. The statistics of time-series precipitation and the streamflow among 125 catchments. Missing rate as limitation.**

|  | Annual Total Precipitation (mm) | Max. Hourly Precipitation (mm) | Annual Mean Streamflow (m3/s) | Missing Rate of Precipitation (Raw Data) | Missing Rate of Streamflow (Raw Data) |
|---|---|---|---|---|---|
| Min | 794 | 9.1 | 3 | 0.02% | 0.69% |
| Max | 1,056 | 60.0 | 12,963 | 0.04% | 33.14% |
| Mean | 952 | 24.8 | 1,926 | 0.02% | 15.16% |
| Median | 961 | 22.2 | 608 | 0.02% | 16.14% |
| SD | 57 | 10.3 | 2,864 | 0.01% | 6.4% |

From the tables above, it is shown that our dataset is limited to a certain range of precipitation since it contains the catchments only in Iowa. As it is shown in Table 3 that all 125 catchments share similar precipitation ranges from 794 to 1056, with a small standard deviation of 57. Geologically, all the catchments are located in two HUC watersheds, the Upper Mississippi and Missouri, and the study results may not be applicable to other regions in the U.S. WaterBench is also subject to a relatively high missing data rate for streamflow since the reliable hourly dataset is limited in USGS for some of the watersheds in Iowa.

In the following sections, we will discuss the details of specific datasets and features.

### 2.2.1 Area

From the water cycle, the precipitation is the main driving force of the streamflow. Based on the 90m digital elevation model (DEM), only the precipitation in a certain area will contribute to a stream. Each measuring station has its corresponding area, which can be calculated from the watershed boundary shapefiles. Since the total precipitation amount is the product of

precipitation intensity and area, larger watersheds typically have higher streamflow rates. In WaterBench, the boundary shapefiles of each watershed are obtained from the Iowa Flood Information System (IFIS), a system operated by the Iowa Flood Center (IFC). Moreover, the area is calculated from the shapefiles in the unit of square kilometers. Thus, the area contains one value per station, and it is available in the column of "area" in the "{station_id}_data.csv" files.

### 2.2.2 Time of Concentration

The time of concentration provides the dimension of stream length for a watershed. In WaterBench, the time of concentration is defined as the longest length over the velocity, which is the time the water concentrates from the most distant point from the watershed outlet. The velocity used in this study is a constant value of 0.75 m/s, which was found appropriate for Iowa basins (Mandapaka et al., 2009; Mantilla et al., 2011), and has been successfully used in many hydrologic models (Fonley et al., 2016; Sloan et al., 2017). Thus, for a long and narrow watershed, it may have a small watershed area but a large time of concentration.

In WaterBench, the time of concentration is obtained from the IFIS with the unit of hours. Thus, the time of concentration contains one value per station, and it is available in the column of "travel_time" in the "{station_id}_data.csv" files.

### 2.2.3 Slope

The Slope is one of the topographic features that represents the slope gradient in percentage. A steep slope may cause a higher velocity and lower infiltration rate, which normally causes a larger streamflow rate at a precipitation event. The original file,

hillslope map, is calculated by IFC (Sit et al., 2019), which split the land of Iowa into over 600,000 hydrologic units using the algorithm developed by Mantilla and Gupta (2005). In WaterBench, the average slope is calculated from the mean value of the hillslopes in each catchment (Gericke and Du, 2012). Thus, the slope is a constant value per watershed, and it is available in the column of "slope" in the "{station_id}_data.csv" files.



### 2.2.4 Soil Type

Soil type is one of the topographic features that represents the proportions of 12 different soil types of the land. Normally, the sandy soil has the largest infiltration rate, and the clay has the least infiltration rate. The original file, global soil types, is available from NASA (Post et al., 2000). It is a 2-D map with a spatial resolution of 0.5 degrees. The soil type proportion is then calculated using the weighted average for each watershed. It needs attention that four dominant soil types, including the loam, silt, sandy clay loam, and silty clay loam, contribute to 99.91% of the area in Iowa. Thus, only these four soil types are

considered in the dataset. The percentage of each soil type is constant in the time series dataset for each station in the column of "loam", "silt", "sandy_clay_loam", and "silty_clay_loam" in the "{station_id}_data.csv" files.

### 2.2.5 Streamflow Rate

The streamflow rate is a variable measured by USGS in the unit of cubic feet per second. The data were acquired from the USGS National Water Information System. There are nearly 200 real-time streamflow measuring stations in Iowa. After

removing the stations established after 2011 or permanently closed before 2018, a total of 125 stations are selected, as shown in Figure 1. These streamflow data were aggregated hourly for each station first. Since there were a few missing values in the original data caused by station system breakdown or internet outages. For the stations located in the northern part of Iowa, the river may freeze and have no flow rate measurement over the winter, and all missing values were reported as -9999 from USGS. In the dataset, each watershed has two columns, with the first column represent timestamp from 2011/10/01 00:00 to

2018/9/30 23:00, and the second column represent the streamflow values. Thus, the streamflow rate contains 61,368 values per station, and they are available in the column of "discharge" in the "{station_id}_data.csv" files.

### 2.2.6 Precipitation Volume

The precipitation volume is a feature that represents how much water is introduced into the watershed from the precipitation. Many station-based and satellite datasets have been measuring precipitation over the years. After comparisons, it is found that

NOAA's Stage IV multi-sensor measurement is the most accurate (Seo et al., 2018) in the State of Iowa. The Stage IV multi-sensor provides the hourly precipitation amount in a 4km-grid spatial resolution. The catchment level average precipitation is then calculated at each hour. Since there is no rainfall or snowfall most of the time, most precipitation values in the dataset are 0. In the dataset, each watershed has two columns, with the first column represents timestamp from 2011/10/01 00:00 to 2018/9/30 23:00, and the second column represents the precipitation on the watershed. Thus, the precipitation data contains

61,368 values per station, and they are available in the column of "precipitation" in the "{station_id}_data.csv" files.

### 2.2.7 Evapotranspiration (ET)

ET represents the evaporation and the plant transpiration from the land in the water cycle. It is one of the major losses of precipitation. Since there is no high-resolution ET dataset available, we used the monthly estimation from the historical





measurement data in the past decades (Krajewski et al., 2017) as an empirical dataset. This is a monthly-based dataset for the

entire state of Iowa. In the dataset, we applied the ET value for each timestamp from 2011/10/01 00:00 to 2018/9/30 23:00.

Thus, the ET data contains 61,368 values for all stations, and they are available in the column of "et" in the

"{station_id}_data.csv" files.

**2.2.8 Watershed Relationship**

Since many USGS measurement gauges are in the same watershed, many catchments in WaterBench are not independent, and

a relation tree is given in the "catchment_relationship.csv". The csv file represents a disconnected directed graph with each

row representing an edge. 63 out of 125 catchments have one or more upstream, as shown in the relationship, which are

relatively large catchments. The remaining 62 catchments are specified as the very upstream catchments which have only one

stream gage. Since these catchments have no overlapping area, the catchments in our dataset form a disconnected graph. For

the catchments have overlapping areas, the watershed ID 646 has the largest connected subgraph with 27 upstream catchments.

With upstream-downstream relationships, WaterBench supports the cutting-edge studies such as graph neural networks.

**3. Benchmark Tasks and Metrics**

In this section we define a sample benchmark task of predicting the hourly streamflow for the next five days for future

comparative studies. In this task, we ignore the errors of the rainfall forecast, and use all the data, including the topology data,

past three days' precipitation and streamflow data, and the future five days' precipitation data as input, to predict the streamflow

for the next 120 hours at the watershed outlet. We take two separate approaches to tackle this problem. The first approach

involves a separate machine learning model for each of the available watersheds while the second one is to build single large

regional model that carry out the same task for all available watersheds.

For this specific task, we selected the last water year as the test set, and the rest as the training set. We further formatted the

original dataset into a ready-to-use structure for each watershed with four files named as train_x, train_y, test_x, test_y. Thus,

totally 500 files for 125 watersheds are provided for this specific task. Since general statistics such as mean squared error

(MSE) and root mean squared error (RMSE) are not dimensionless, the metrics for this study are Nash-Sutcliffe efficiency

(NSE) and Kling-Gupta efficiency (KGE). They both are dimensionless statistics that widely used in the hydrological studies,

and can be used to compare between watersheds. Both NSE and KGE range from negative infinity to 1, and the closer to 1 the

better. The equations 1 and 2 for NSE and KGE are shown below:

$\text{NSE} = 1 - \frac{\sum_{i=1}^{n}(Y_i - \hat{Y}_i)^2}{\sum_{i=1}^{n}(Y_i - \bar{Y})^2}$                                            Eq. 1

$\text{KGE} = 1 - \sqrt{(r-1)^2 + (\frac{\sigma_{\hat{Y}_i}}{\sigma_{Y_i}} - 1)^2 + (\frac{\mu_{\hat{Y}_i}}{\bar{Y}} - 1)^2}$          Eq. 2

where: $Y_i$ is the observation at the time $i$; $\hat{Y}_i$ is the model result at the time $i$; $\bar{Y}_i$ is the mean of all observations; $n$ is the total number of observations; r is the Pearson correlation coefficient; $\sigma$ is the standard deviation; and $\mu$ is the mean.

Both NSE and KGE are dimensionless and in the range of (-∞,1]. For both metrics, the closer to 1, the better model performs.

We calculate the NSE and KGE based on the test year for each prediction hour. This means that there will be 120 different NSE and KGE values for different hours at each watershed. It should be noted that since the watersheds here are not filtered, it is possible for some watersheds to be greatly affected by human activities, including mitigation, construction, irrigation, urban drainage, etc. activities in watersheds. Thus, a median value of all 125 watersheds is meaningful to report as a widely employed practice within other hydrology studies (Kratzert et al., 2018, Xiang et al., 2020). In addition, since the prediction

accuracies typically decrease when the lead time increases, the median NSE and KGE of 125 stations at the 120hr ahead predictions is the most important value to report.

## 4. Benchmark Results and Discussion

To provide baseline results over the sample benchmark task and two approaches defined in the previous section, we employed a linear regression model using Ridge regression, and three deep learning models using LSTM, GRU, and sequence-to-

sequence (S2S) network architectures. For the first approach, we considered each watershed independent and trained one model for each watershed. Thus, relationship between the watersheds are not used in this benchmark. The median NSE and KGE scores among 125 watersheds at each hour are shown in Figure 3 and Table 4. As shown in the figure and the table, the Ridge regression has a high accuracy at the first 24 hours since the streamflow rates normally do not change too much in one day, and they are relatively easy to predict. The metrics for the long term show that the model using GRU has the best

performance. The NSE and KGE histogram of GRU shows that for most of the watersheds GRU model performs well and only in a limited number of watersheds the GRU model gives negative scores.

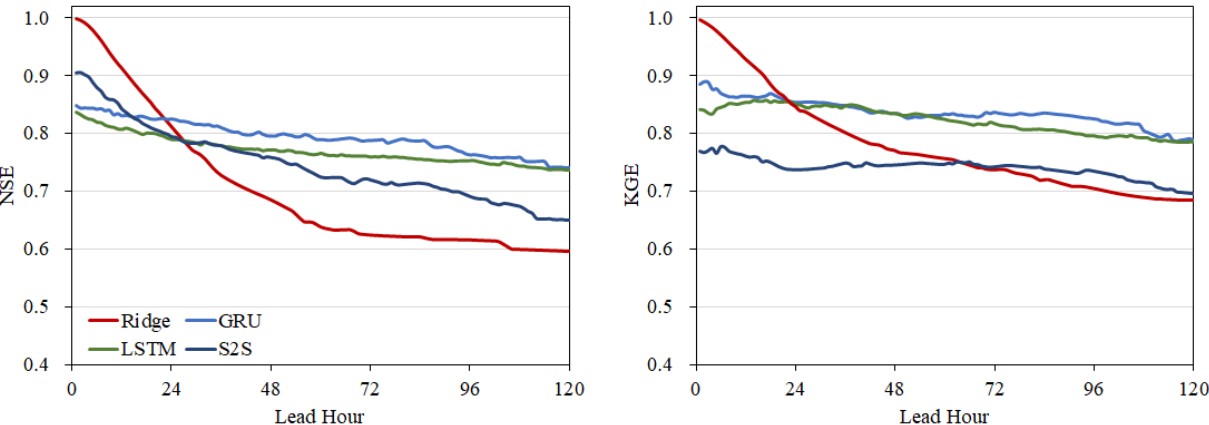

**Figure 3. The median NSE and KGE among 125 watersheds in 125 different models at the prediction of the next 1 to 120 hours.**



**Table 4. The median NSE and KGE among 125 watersheds at the prediction hour 1, 6, 12, 24, 48, 72, 96, and 120.**

| | | | NSE | | | KGE | | |
|---|---|---|---|---|---|---|---|---|
| **Hour** | **Ridge** | **GRU** | **LSTM** | **S2S** | **Ridge** | **GRU** | **LSTM** | **S2S** |
| 1 | **0.998** | 0.848 | 0.836 | 0.905 | **0.996** | 0.884 | 0.840 | 0.769 |
| 6 | **0.970** | 0.841 | 0.819 | 0.879 | **0.970** | 0.868 | 0.844 | 0.777 |
| 12 | **0.912** | 0.830 | 0.806 | 0.842 | **0.928** | 0.863 | 0.853 | 0.761 |
| 24 | 0.811 | **0.825** | 0.789 | 0.794 | 0.847 | **0.853** | 0.849 | 0.738 |
| 48 | 0.685 | **0.795** | 0.771 | 0.758 | 0.771 | **0.833** | 0.834 | 0.746 |
| 72 | 0.624 | **0.787** | 0.759 | 0.719 | 0.738 | **0.836** | 0.817 | 0.742 |
| 96 | 0.616 | **0.762** | 0.753 | 0.691 | 0.705 | **0.824** | 0.796 | 0.735 |
| 120 | 0.596 | **0.740** | 0.736 | 0.649 | 0.685 | **0.787** | 0.786 | 0.696 |

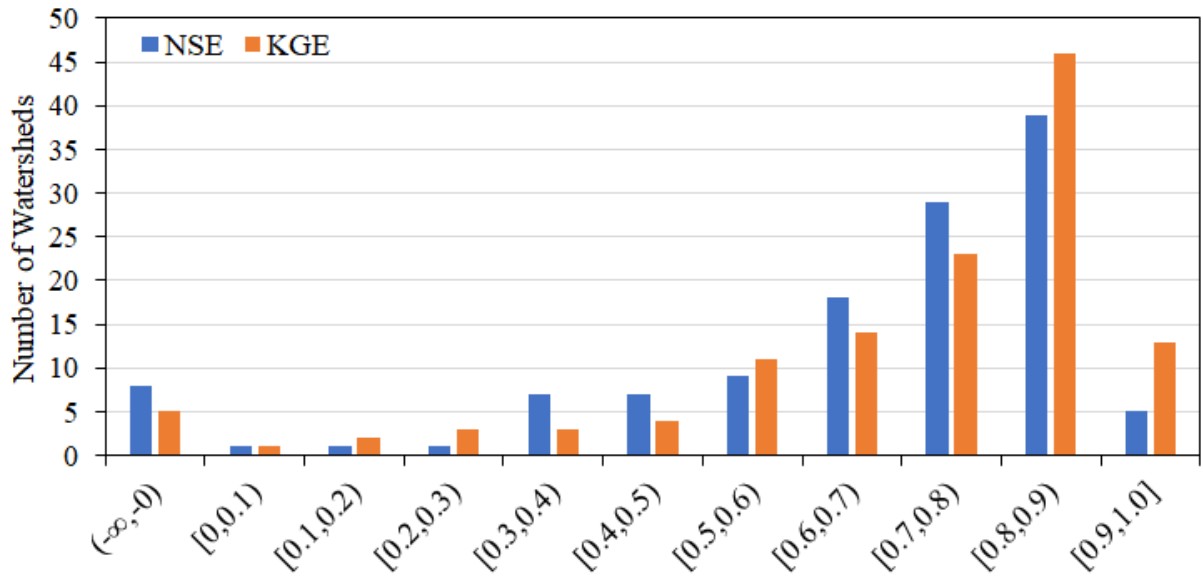

**Figure 4. Histogram of the GRU model performance.**



As for the second approach, we attempted to develop single regional model for all 125 watersheds since they share similar physical attributes. As shown in Figure 5, a single model on all 125 watersheds is possible with the physical features including area, slope, travel time, and soil types using the customized NSE loss function (Xiang et al., 2021). Among four models, similar to the first approach, the performance of Ridge regression is hard to beat at first. Nevertheless, the deep learning model S2S starts to show a better performance starting the second day.

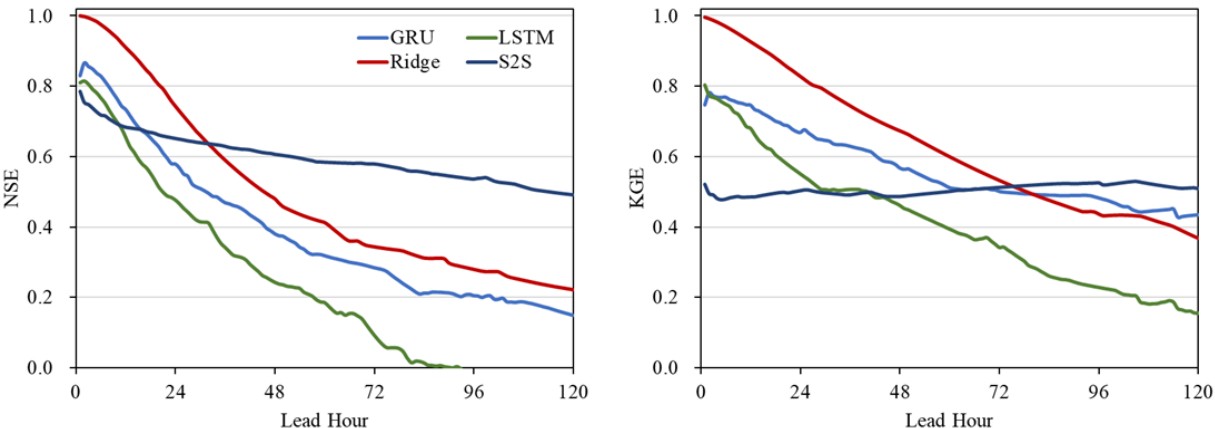


**Figure 5. The median NSE and KGE among 125 watersheds using single regional model at the prediction of the next 1 to 120 hours.**

**Table 5. The median NSE and KGE among 125 watersheds at the prediction hour 1, 6, 12, 24, 48, 72, 96, and 120.**

|  |  |  | NSE |  |  | KGE |  |  |
|---|---|---|---|---|---|---|---|---|
| Hour | Ridge | GRU | LSTM | S2S | Ridge | GRU | LSTM | S2S |
| 1 | **0.999** | 0.831 | 0.809 | 0.785 | **0.996** | 0.747 | 0.804 | 0.522 |
| 6 | **0.974** | 0.829 | 0.766 | 0.718 | **0.971** | 0.769 | 0.749 | 0.479 |
| 12 | **0.910** | 0.734 | 0.654 | 0.683 | **0.927** | 0.746 | 0.681 | 0.485 |
| 24 | **0.743** | 0.579 | 0.476 | 0.651 | **0.828** | 0.668 | 0.551 | 0.503 |
| 48 | 0.480 | 0.381 | 0.243 | **0.606** | **0.673** | 0.567 | 0.462 | 0.487 |
| 72 | 0.343 | 0.285 | 0.093 | **0.579** | **0.532** | 0.500 | 0.342 | 0.513 |
| 96 | 0.279 | 0.205 | -0.042 | **0.535** | 0.437 | 0.482 | 0.228 | **0.526** |
| 120 | 0.221 | 0.149 | -0.241 | **0.491** | 0.368 | 0.434 | 0.155 | **0.509** |

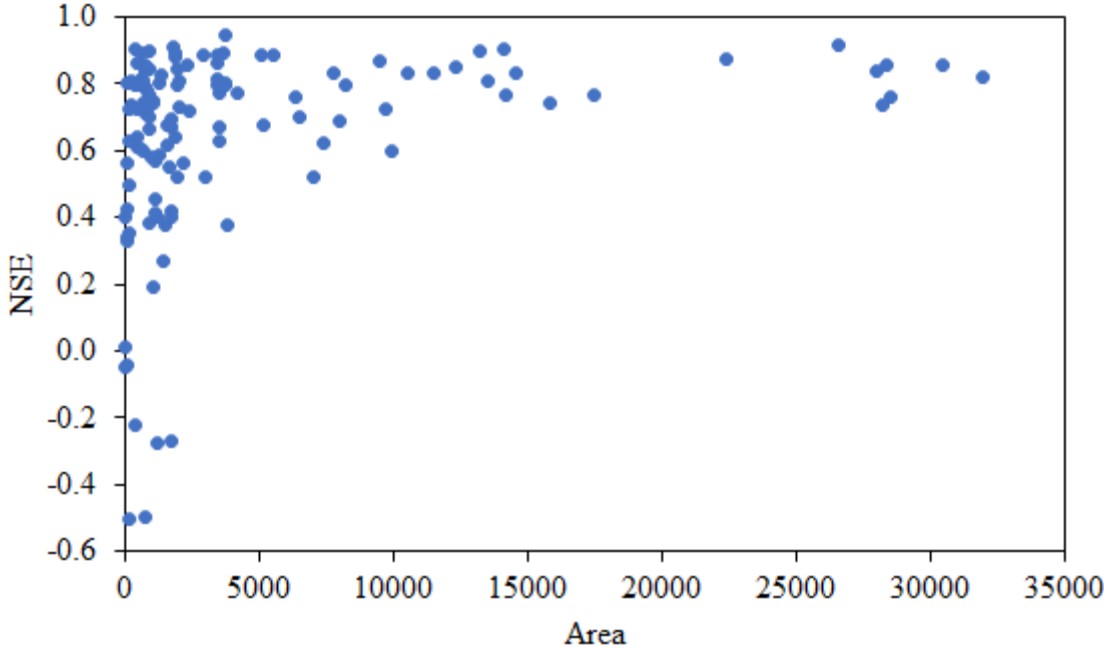


**Figure 6. The distribution of the 120 hours ahead prediction using the best model in our benchmark (GRU for the single station).**

As shown in the results, there are two major limitations. First, the model efficiency is low on the first day. It is shown in Figure 3 and Table 4 that the deep learning models do not show a higher accuracy at the first several hours compared to the Ridge

model. Some hydrological studies have also shown that the basic persistence model (Streamflow t+n = Streamflow t) is a hard-to-beat for short-range predictions when n is smaller than 12 hours (Krajewski et al., 2020). Thus, it is hard to make both short-range and long-range predictions accurate in one model. The second limitation is the scale effect. The results show that as watersheds get larger, the predictions become easier and better. This means the small watersheds, typically representing the middle and upper reaches, are harder to predict. Figure 6 shows the drainage area and 120-hr ahead prediction performance in

NSE for 125 watersheds. The scale effect observed in our benchmark indicates the prediction on small watersheds is still a challenge.

Although a lot of metadata is provided in our dataset, as a benchmark, our study does not consider complex pretreatment nor models with domain knowledge in hydrology. Some recent studies have shown that the moving average for smoothing, the consideration of time lag, the consideration of watershed upstream-downstream connections, and other deep learning model

architectures may be effective for a better prediction. However, these studies are based on their own dataset, and the results cannot be directly compared. We encourage researchers to conduct comparisons based on the WaterBench.





## 5. Conclusion

In this study, by aggregating the datasets of watershed area, slope, soil types, streamflow, precipitation, and ET from NASA, NOAA, USGS, and IFC, we presented a dataset, namely WaterBench, that is prepared for an hourly streamflow forecast task. This dataset has a high temporal resolution with abundant geographic and relational information, which can be used for varieties of deep learning and machine learning applications research. We defined a sample streamflow forecasting task for the next 120 hours and provided example benchmark results on this task with a traditional linear and three custom deep learning models.

WaterBench is not filtered and thus represents an actual streamflow forecast problem as much as possible. Although the data is limited to the Midwest, we believe that any studies on this dataset could provide insights for other streamflow forecasting and rainfall-runoff modeling studies at other watersheds. With the open-source release of WaterBench (https://github.com/uihilab/WaterBench), this work provides a comparable benchmark, which to some extent makes up for the lack of a unified benchmark in hydrological and water resources research. We highly encourage other researchers to use the WaterBench in their hydrological modeling research studies.

## 6. Data and Code Availability

The data and codes that support this study are openly available in our open-source GitHub repository at https://github.com/uihilab/WaterBench (Demir et al., 2022). The dataset covers the 125 catchments in Iowa, U.S. with seven different features, including precipitation, streamflow rate and ET with available data from October 1st, 2011 (the water year 2012) to September 30th, 2018 (the water year 2018). The original files of the dataset, metadata, and sample codes can be downloaded from archive files. Four different models including Ridge, LSTM, GRU, S2S, used in our paper are provided with ready to run Python Jupyter Notebooks as well (Demir et al., 2022). It is welcome to send us feedback by filing an issue on the repository.

## 7. Acknowledgments

The work reported in this study was made possible by the support of members of the Iowa Flood Center and the Department of Civil and Environmental Engineering at the University of Iowa.

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
