# Peer review of "WaterBench-Iowa: A Large-scale Benchmark Dataset for Data-Driven Streamflow Forecasting"

_Earth System Science Data, 2022_

## Referee Comment (RC2)

**Review of ESSD-2022-52: WaterBench: A Large-scale Benchmark Dataset for Data-Driven Streamflow Forecasting**

In their manuscript, the authors present a hydrological benchmark product particularly for ML-approaches. They've collected hydrological time-series data for precipitation, streamflow, evapotranspiration as well as static data for soil types, slopes, etc. mainly for the state of Iowa. This data is area-averaged over several small basins and arranged in a way, that each of their files contains all data for a single sub-basin. They also include a file that describes the relationships between the different sub-basins.

Using this data, they compute and evaluate some 5-day-forecasts by applying several ML-approaches as well as Ridge regression.

Overall, I found the idea very interesting and such a well-curated and put-together dataset would be an important contribution to both the hydrological and ML-community; there would also be a lot of potential for enhancing such a product with more variables (e.g., temperature, wind, etc.) and also include other states beyond Iowa.

However, after going through the manuscript and looking at the data, I have several points of criticism. Thus, although I really liked the general idea and also acknowledge the authors motivation to make all data and codes publicly available, I think that the paper, as well as the dataset in its current form, needs some substantial revision. Thus, I suggest to reject the manuscript but also encourage the authors to re-submit a revised version.

**Major comments**

- My main concern is that the focus of the paper is not clear. If the authors want to present a state-of-the-art "meteorological forcing" product for testing ML-approaches (which would be very interesting), then I do not understand the choice of datasets (see below) and the structure of the dataset needs some revision (probably as one large NetCDF?). It would also be desirable to provide some alternative data (e.g., different soil maps, different precipitation data, etc.) in order to run, e.g., ensemble experiments. However, if the authors rather focus on the development of ML-based streamflow reference forecasts, against which other ML-approaches can be tested, then the methods-section as well as the uncertainty-analysis needs to be substantially enhanced. And, even if the authors claim several times that flood forecasting is an important task (which requires very detailed and precise evaluations), the results-section is restricted to a very high-level comparison of median NSE and KGE values across all the 125 stations.

- Overall, the whole presentation of the ML-predictions is lacking a lot of details. Thus, even if the authors state that their predictions are just examples what one could do with their dataset, there is basically no methods-section and no discussion, why the chosen ML models might be appropriate for flood forecasting. Instead, this whole experiment seems to be another example of a more or less loose conglomerate of ML-approaches. Due to the extreme frequency where new and even more user-friendly ML-libraries are released practically every day, there are more and more papers where people simply use these methods because they can! Here, the authors simply state that they're using Ridge

regression, LSTM, GRU and S2S. It remains unclear why (or why not) these approaches are particular suited for this application and also the reasons for the differences are left completely open.

- Dataset / manuscript currently does not comply with ESSD-regulations:
    - Your dataset needs a DOI (https://www.earth-system-science-data.net/policies/data_policy.html)...
    - ...which you can obtain by uploading to a long-term repository (https://www.earth-system-science-data.net/policies/repository_criteria.html
    - This DOI should also be added to your abstract (https://www.earth-system-science-data.net/submission.html)
    - Furthermore, while it is highly acknowledged that the authors have made all their code and data openly available, I found especially the structure of the data very hard to understand!

**Minor comments**

- The authors could have chosen more appropriate datasets for the different water-cycle variables (or even provide an ensemble). Especially as the authors apply area-aggregated precipitation and evapotranspiration, there is a huge range of products available. Instead, the authors mix hourly precipitation with monthly evapotranspiration as well as high- with low-resolution data and, hence, provide a quite inconsistent product with a lot of room for improvement.

- The authors cite that "deep neural networks increased scientists' ability in modelling both linear and non-linear problems without time-intensive data engineering processes by domain experts". At another section, they also claim that the application of a particular dataset does not require domain knowledge from meteorologists (line 85). To be honest, I do not think that this is actually a desirable development. I even think that some proof-reading from a "domain native" could have substantially improved the paper as the authors have, at several places, chosen some quite strange and uncommon wording (see particularly the description of the different variables section 2; some examples are given below).

- It is also not clear how the authors defined their benchmark setup. I assume that they made 5-day-predictions on every day during their test year. From these predictions, they combined all forecasts that refer to a specific lead time (e.g., 1 hour, 6 hours, etc.) into a single sample, over which they then compute the NSEs and KGEs. In their figures and tables, they, finally, present the median NSE and KGE values over all 125 basins. Is that, more or less, correct?

- Lines 33 - 34: I am not sure if "flood forecasting" is really a synonym for "streamflow prediction / runoff forecasting". Furthermore, "runoff" usually describes water on an area that does not infiltrate or evaporate and, hence, discharges from that area; streamflow is the actual discharge that you measure in a river or channel. Thus, while these terms are very related, they do not mean the same thing.

- Line 69: I found the wording here quite strange. You usually do not forecast measurements, but some phenomenon like precipitation or runoff.

- Line 120: "The WaterBench is not selected based on human activities, which is a reaction to the real situation in Iowa" --> What do you want to say here?

- Lines 139 - 144: This sounds not very "hydrological". You should rather say that red dots are located at outlets of larger basins, which are divided into several smaller upstream sub-basins. And the outflow from each sub-basin is measured at the green gauging stations. See, e.g., http://proceedings.esri.com/library/userconf/proc01/professional/papers/pap1008/p100 8.html for a good explanation of the terminology. For better visibility, it might make more sense to separate the larger basins from their sub-basins, e.g., by using thick and thin lines in Figure 1.

- Line 148: What are "statistics of the metadata"?

- Line 149 - 150: "Metadata" usually refers to "data that provides information about data". You mean simply "static data". Please rephrase: For each basin, we provide static data (area, slope, travel time, ...) as well as time-series for streamflow, precipitation, and ET.

- Line 195: Your particular soil dataset has 12 soil-types. And I am pretty sure, that there are newer and much higher resolved maps available (see, e.g., https://www.cen.uni-hamburg.de/icdc/data/land/soilmap.html). Such data would fit better to your high-resolved precipitation data (even if you're only looking at basin averages).

- Line 206: Please re-phrase: For each station, streamflow data was aggregated to hourly values.

- Line 213: This sounds, once more, very "un-hydrological" and I strongly assume that you don't have to explain what "precipitation" is.

- Lines 206 - 207: Since there were... -> Please check your grammar... This sentence does not make sense.

- Lines 218 - 220: This sounds, again, overly complex... For "precipitation on the watershed", you usually say "basin-averaged precipitation".

- Lines 222 - 223: Describing evapotranspiration as a major loss of precipitation sounds quite uncommon. Maybe say "precipitated water".

- Line 223: "no high-resolution ET dataset" → What about ERA5 (Land)? GLEAM? MERRA? MSWX?

- Line 224: What do you mean with "empirical dataset"?

- Line 252: Please rephrase: where Y_i is the observation at time i, Y^hat_i is the model result at time i, ...; and please add that sigma and mu refer to the forecasts while your Y refer to the observations! Furthermore, you usually use the same parameter "family" for the mean and standard deviation. So, either use Greek letters (as, e.g., in

https://hess.copernicus.org/preprints/hess-2019-327/hess-2019-327.pdf), or stick to Latin letters (as in your Equation 1).

- Line 258: "Thus, a median value...": While this is certainly true, it also makes a lot of sense to analyse the distribution of your performance metrics across your 125 basins in order to get an idea where and why your model performs better/worse. Figure 4 is only a starting point for such an analysis.

- Line 260: Why are the 120hr ahead predictions the most important values?

- Line 297: I would consider 5-day-forecasts as "medium-range

- Figure 2: Presenting CDFs for static parameters is quite unusual. It would be more helpful if you show Histograms here.

- Table 3: You would rather say "Summary statistics for precipitation and streamflow". And, for better readability, please add the period during which these numbers were calculated.

---

## Referee Comment (RC3)

**General comments**

This paper proposes a comprehensive reference dataset for streamflow prediction based on use in data-driven and machine learning studies, providing reference performance for more advanced deep learning architectures on datasets for comparative analysis. There is certainly a lack of unified reference data in earth science research, and these studies are very useful in being able to outline a general algorithm for defining them. A detailed description of the results is given in the study, but a more detailed description on the data, but also on the climatology of the study area should probably be reported to provide more details for researchers who intend to use WaterBench for deep learning research in hydrology.

**Specific comments**

Line 17: *"To some extent, WaterBench makes up for the lack of unified benchmarks in earth science research. We highly encourage researchers to use the WaterBench for deep learning research in hydrology."*
**Suggestion**: I would replace this sentence with a summary of the results; it will be the good performance that will encourage.

Line 84: *"This dataset solves the difficulty of data acquisition and does not require domain knowledge from meteorology."*
**Question**: What the authors mean by this sentence: *"… does not require knowledge of the domain from meteorology?"*

Line 89: *"Scientific advancement, intrinsically, is supposed to be cumulative, and in order to have better generalized deep learning-based flood forecasting models, scientists need to build on top of what their fellow researchers have done."*
**Comment**: Certainly, scientists need to build on what their fellow researchers have done, but they also have to improve on what has been done.

Line 90: *"We believe that this could only be done by using the same set of testing mechanisms …"*
**Comment**: This sentence is true after being certain that everything has been done correctly.

Line 102: *"There remains a need for a dataset that is more convenient to use in deep learning research given that most of the deep learning researchers are not domain experts."*
**Question**: What the authors mean by this sentence?

Line 110: *"This study proposes a flood forecasting dataset that follows FAIR data principles …"*
**Suggestion**: I would write some references and details about FAIR.

Line 115: *"… it could be used in other studies with similar goals, such as physically based modeling with physical equations."*
**Suggestion**: I think "*with physical equations*" is not needed.

Line 165: *"From the tables above, it is shown that our dataset is limited to a certain range of precipitation since it contains the catchments only in Iowa."*
**Question**: What the authors mean by this sentence?

Line 167: *"Geologically, all the catchments are located in two HUC watersheds, the Upper Mississippi and Missouri, and the study results may not be applicable to other regions in the U.S."*
**Question**: Do the authors think that this study is only about the analyzed area?

Line 168: *"WaterBench is also subject to a relatively high missing data rate for streamflow since the reliable hourly dataset is limited in USGS for some of the watersheds in Iowa."*
**Question**: How have you thought about solving this limitation? How much data is missing?

Line 174: *"Since the total precipitation amount is the product of precipitation intensity and area, larger watersheds typically have higher streamflow rates."*
**Comment**: I do not think this comment is needed, because river flow discharge depend on a number of conditions.

Line 179: *"2.2.2 Time of Concentration."*
**Question**: Isn't the time of concentration calculated for each grid point of the considered domain? Is the velocity not estimated as a function of slope?

Line 188: *"A steep slope may cause a higher velocity and lower infiltration rate, which normally causes a larger streamflow rate at a precipitation event"*
**Suggestion**: Therefore, it would be very important to have the velocity data for the different considered grid points.

Line 206: *"Since there were a few missing values in the original data caused by station system breakdown or internet outages."*
**Question**: How is missing data handled?

Line 221: *"2.2.7 Evapotranspiration (ET)."*
**Question**: Do the authors believe that this comprehensive reference dataset for streamflow prediction is intended for climate or meteorological studies?
For climate studies the monthly ET parameter may be useful; for meteorological-hydro studies perhaps, it would be more functional to use soil moisture data (also useful for climate studies) and at a higher temporal resolution.

Line 145: *"Since general statistics such as mean squared error (MSE) and root mean squared error (RMSE) are not dimensionless…"*
**Suggestion**: However, it would be interesting to see some results.

Line 255: "*This means that there will be 120 different NSE and KGE values for different hours at each watershed*."
**Question**: What the authors mean by this sentence?

Line 256: "*It should be noted that since the watersheds here are not filtered, it is possible for some watersheds to be greatly affected by human activities, including mitigation, construction, irrigation, urban drainage, etc. activities in watersheds.*"
**Question**: Doesn't this sentence conflict with what is stated in line 120?

Line 275: "Table 4. The median NSE and KGE among 125 watersheds at the prediction hour 1, 6, 12, 24, 48, 72, 96, and 120."
**Question**: what do the second and third columns indicate?

Line 287: "Table 5. The median NSE and KGE among 125 watersheds at the prediction hour 1, 6, 12, 24, 48, 72, 96, and 120."
**Question**: what do the second and third columns indicate?

Line 297: *"The second limitation is the scale effect."*
**Comment**: Unfortunately, it is not clear from the work what scales are being considered. Would the scale effects be the size of the drainage basin?

Line 300: "The scale effect observed in our benchmark indicates the prediction on small watersheds is still a challenge."
**Question**: Given the Classification of Watersheds by Size from the values in Table 2 it is evident the small to medium sized watersheds in the considered domain are a large number. I do not understand the statement made in line 300.

Line 315: *"Although the data is limited to the Midwest, we believe that any studies on this dataset could provide insights for other streamflow forecasting and rainfall-runoff modeling studies at other watersheds."*
**Comment**: This sentence written in this way is contrary to what is stated in line 168.

---

## Author Response (AR1)

**WaterBench: A Large-scale Benchmark Dataset for Data-Driven Streamflow Forecasting**

**Response to Reviewers' Comments**

ESSD-2022-52 | Data description paper
Submitted on 09 Feb 2022
Ibrahim Demir, Zhongrun Xiang, Bekir Demiray, and Muhammed Sit
Special Issue: Benchmark datasets and machine learning algorithms for Earth system science data (ESSD/GMD inter-journal SI)

**Editor:**

Dear authors,

all three reviewers see substantial value in the benchmark dataset you have put together. However, especially reviewer 2 raises some major concerns regarding the dataset and benchmark description. I second her/his concerns and would like to remind you that the definition of an ML benchmark involves a clear problem statement, useful and universally accepted validation metrics and a "vanilla solution" besides the dataset itself. When answering the reviews and preparing a revised manuscript version, please extend the manuscript title to alert readers to the regional limitations of the dataset you provide.

Thank you, Martin Schultz (editor)

**Answer:** Thank you for your time and we have updated the manuscript with a more clear benchmark setup, metrics, and more analyses on benchmark model results.

**Community#1:**

Brief comment: I applaud the authors for putting together this useful dataset and manuscript.

**Suggestions:**

1. Without going into depth with its content, I highly encourage the authors to mention in the title and abstract that this dataset is entirely within Iowa. This would help readers understand its content from the get-go. In addition, this would allow for future WaterBench versions in other regions (just like CAMEL; this could be renamed WaterBench-Iowa or WaterBench-IA).

**Answer:** Thank you for the comment. We have updated the title and abstract to WaterBench-Iowa.

**Reviewer#1:**

This paper describes a dataset with time series of 125 monitored catchments and some examples of how it can be used to evaluate forecasting models. Science is a cumulative effort and this dataset fills a gap in the publicly available to hydrology and AI community. The dataset itself is open and in a good condition. It would be nice to have the watershed boundaries and coordinates of each measuring point as shapefiles available together with the data. However, this is not a major concern since they are available from Iowa Flood Center.

I do not see any major flaws In this paper and recommend that it gets accepted

**Specific comments Individual scientific questions/issues**
Open data is important to the scientific community and I welcome this contribution by the authors.

**Technical corrections:**

1.  I am a non-native English speaker and struggled with this sentence on row 25 and it could be rephrased for clarity. This is the sentence in question:

    "The power of deep learning in problem-solving has opened ways to advancements in many fields that machine learning has been a go-to solution for predictive modeling, such as image recognition and synthesis (Demiray et al., 2021), speech recognition, language modeling, and time-series prediction."

**Answer:** We have rephased this sentence to "Deep learning's predictive modeling capabilities have led to improvements in various fields, including image recognition and synthesis (Demiray et al., 2021), speech recognition, language modeling, and time-series prediction."

**Reviewer#2:**

In their manuscript, the authors present a hydrological benchmark product particularly for ML-approaches. They've collected hydrological time-series data for precipitation, streamflow, evapotranspiration as well as static data for soil types, slopes, etc. mainly for the state of Iowa. This data is area-averaged over several small basins and arranged in a way, that each of their files contains all data for a single sub-basin. They also include a file that describes the relationships between the different sub-basins.

Using this data, they compute and evaluate some 5-day-forecasts by applying several ML-approaches as well as Ridge regression.

Overall, I found the idea very interesting and such a well-curated and put-together dataset would be an important contribution to both the hydrological and ML-community; there would also be a lot of potential for enhancing such a product with more variables (e.g., temperature, wind, etc.) and also include other states beyond Iowa.

However, after going through the manuscript and looking at the data, I have several points of criticism. Thus, although I really liked the general idea and also acknowledge the authors motivation to make all data and codes publicly available, I think that the paper, as well as the dataset in its current form, needs some substantial revision. Thus, I suggest to reject the manuscript but also encourage the authors to re-submit a revised version.

Dear reviewer. Thank you for your time and your valuable comments!

Other variables such as temperature, wind speed, wind direction, snowfall, and snow cover were applied in our preliminary research. However, those variables did not improve the performance of the model performance of the 120hr streamflow forecast. In contrast, a powerful data-driven model may easily overfit to features that are not statistically associated with short- or medium-range floods. In addition, one of the goals of our dataset is to be used directly by computer scientists or data scientists who are not familiar with the hydrology domain. Thus, we limited our features so that scientists in other domains could also use the dataset. Due to our regional focus on Iowa flood studies in data collection, we have renamed the title of the manuscript to reflect our spatial focus.

For the rest of your comments, we have the reply one by one.

**Major comments:**

1.  My main concern is that the focus of the paper is not clear. If the authors want to present a state-of-the-art "meteorological forcing" product for testing ML-approaches

(which would be very interesting), then I do not understand the choice of datasets (see below) and the structure of the dataset needs some revision (probably as one large NetCDF?). It would also be desirable to provide some alternative data (e.g., different soil maps, different precipitation data, etc.) in order to run, e.g., ensemble experiments. However, if the authors rather focus on the development of ML-based streamflow reference forecasts, against which other ML-approaches can be tested, then the methods-section as well as the uncertainty-analysis needs to be substantially enhanced. And, even if the authors claim several times that flood forecasting is an important task (which requires very detailed and precise evaluations), the results-section is restricted to a very high-level comparison of median NSE and KGE values across all the 125 stations.

**Answer:**

We used the domain knowledge of hydrology to preprocess the hydrological data in various formats (e.g., grib2 and NetCDF), and summarized them in the general data format CSV which is a widely used format by data scientists. We have tried a variety of rainfall products in our preliminary experiments, and this set of data we provided has been proved to be effective in Iowa studies. Our goal is to enable more computer scientists and data scientists who lack the hydrology knowledge to develop data-driven models on our dataset. And this set of ready-to-use data will be helpful for them.

In today's data science fields, there are many competition platforms providing a dataset and evaluation metrics for users to build the model and break the record. And the leaderboard is based on one or more overall scores as the main evaluation metrics. Here are two samples from two famous platforms.

- Study of the impact of air quality on death rates.
https://www.kaggle.com/competitions/predict-impact-of-air-quality-on-death-rates/leaderboard

This is a competition launched by the European Centre for Medium Range Weather Forecasts (ECMWF), 2017. This dataset consists of only several CSV files, which simply provide a region ID, mean O3, PM10, PM2.5, NO2, temperature, and mortality rate in a table. The leaderboard is based on the RMSE score of the predicted mortality rate.

- Study of the prediction of monthly air quality in Beijing.

https://paperswithcode.com/sota/multivariate-time-series-imputation-on

This is a dataset that provides 7 features (wind speed, wind direction, rainfall, air temperature, dew point temperature, and air pressure) as input to predict the air quality. The score of the leadboard is MAE of daily PM2.5 prediction among 12 sites.

NSE and KGE are two of the most widely used performance metrics in physical and data-driven hydrological forecast studies. Thus, we are providing the median NSE and KGE as the main metrics in this benchmark.

2. Overall, the whole presentation of the ML-predictions is lacking a lot of details. Thus, even if the authors state that their predictions are just examples what one could do with their dataset, there is basically no methods-section and no discussion, why the chosen ML models might be appropriate for flood forecasting. Instead, this whole experiment seems to be another example of a more or less loose conglomerate of ML-approaches. Due to the extreme frequency where new and even more user-friendly ML-libraries are released practically every day, there are more and more papers where people simply use these methods because they can! Here, the authors simply state that they're using Ridge regression, LSTM, GRU and S2S. It remains unclear why (or why not) these approaches are particular suited for this application and also the reasons for the differences are left completely open.

**Answer:** We have added a new section to discuss why these approaches are particularly suited for this application. The LSTM, GRU, and Sequence-to-sequence models on the runoff prediction and streamflow forecast have been widely studied in recent years.

*To sum up, deep learning models such as LSTM have been used in meteorology and hydrology studies of 80 soil moisture modeling (Fang et al., 2017), water table depth prediction (Zhang et al., 2018), rainfall-runoff modeling (Hu et al., 2018; Kratzert et al., 2018), streamflow forecasting (Xiang et al., 2020), etc. As is presented by perspective studies (Reichstein et al. 2019), deep learning models such as LSTM can extract spatial-temporal features automatically to gain further process understanding of Earth system science problems. Therefore, we pay great attention to the application of LSTM and its variant models in this research.*

3. Dataset/manuscript currently does not comply with ESSD-regulations:
    3.1. Your dataset needs a DOI (https://www.earth-system-science-data.net/policies/data_policy.html)...
    3.2. ...which you can obtain by uploading to a long-term repository (https://www.earth- system-science-data.net/policies/repository_criteria.html)
    3.3. This DOI should also be added to your abstract (https://www.earth-system-science- data.net/submission.html)

3.4.  Furthermore, while it is highly acknowledged that the authors have made all their code and data openly available, I found especially the structure of the data very hard to understand!

**Answer:** For 3.1, 3.2, 3.3, we have submitted the dataset to a long-term repo with a DOI and updated the manuscript. Here it is. https://zenodo.org/record/7011838#.YyQ0FXbMIQ8

The data structure is in CSV files, which is machine learning ready-to-use structure and can be read as a data frame directly. Our paper mainly described the raw data. For the ready-to-use datasets, we suggest reading through the function ***read_file*** in our sample codes.

**Minor comments:**

1.  The authors could have chosen more appropriate datasets for the different water-cycle variables (or even provide an ensemble). Especially as the authors apply area-aggregated precipitation and evapotranspiration, there is a huge range of products available. Instead, the authors mix hourly precipitation with monthly evapotranspiration as well as high- with low-resolution data and, hence, provide a quite inconsistent product with a lot of room for improvement.

**Answer:** We are providing the same spatial resolution as CAMELS (Newman, 2015), and our benchmark mainly focuses on temporal predictions. There is no real-time hourly evapotranspiration observations, so the mixing and efficient use of different resolutions is also a challenge in modeling and could be an advantage for deep learning. We are trying to provide best available datasets for all physical parameters that are available for supporting real-time prediction. Our goal in creating this task is to implement real-time streamflow forecasts that take advantage of the efficient performance of machine learning. Therefore, we only consider the data that can be obtained during real-time forecasting, including monthly ET that can represent the seasonal changes and so on. Many water cycle variables can be obtained from reanalysis data but not real-time. The original data from multiple resources were inconsistent, but this is exactly our work: our aggregated dataset provides a USGS basin station-level hourly data.

Newman, A. J., Clark, M. P., Sampson, K., Wood, A., Hay, L. E., Bock, A., ... & Duan, Q. (2015). Development of a large-sample watershed-scale hydrometeorological data set for the contiguous USA: data set characteristics and assessment of regional variability in hydrologic model performance. *Hydrology and Earth System Sciences*, *19*(1), 209-223.

2.  The authors cite that "deep neural networks increased scientists' ability in modelling both linear and non-linear problems without time-intensive data engineering processes

by domain experts". At another section, they also claim that the application of a particular dataset does not require domain knowledge from meteorologists (line 85). To be honest, I do not think that this is actually a desirable development. I even think that some proof- reading from a "domain native" could have substantially improved the paper as the authors have, at several places, chosen some quite strange and uncommon wording (see particularly the description of the different variables section 2; some examples are given below).

**Answer:** We see this as a new kind of collaboration. The raw data is preprocessed by domain experts for some specific tasks, and the tasks are then converted into solving an optimization problem (i.e., language translation models can be applied to the time-series forecast modeling in earth science studies because they are the same sequence step-by-step forecast task [Figure 2d, Reichstein et al., 2019]). It is not an issue if the dataset is correctly processed by domain experts before handed over to data scientists or computer scientists.

Reichstein, M., Camps-Valls, G., Stevens, B., Jung, M., Denzler, J., & Carvalhais, N. (2019). Deep learning and process understanding for data-driven Earth system science. Nature, 566(7743), 195-204.

3.     It is also not clear how the authors defined their benchmark setup. I assume that they made 5-day-predictions on every day during their test year. From these predictions, they combined all forecasts that refer to a specific lead time (e.g., 1 hour, 6 hours, etc.) into a single sample, over which they then compute the NSEs and KGEs. In their figures and tables, they, finally, present the median NSE and KGE values over all 125 basins. Is that, more or less, correct?

**Answer:** Thank you for the comment. We described the benchmark setup in Line 237-245 in the revised manuscript. Most current studies (i.e., Kratzert et al., 2019) are using median NSE/KGE over hundreds of basins as the main metric. In addition, as is also mentioned in our answer to your Major comment #1, a simple metric for a task is important in today's benchmarking studies. In this revised manuscript, we also included CDF as you mentioned.

Kratzert, F., Klotz, D., Shalev, G., Klambauer, G., Hochreiter, S., & Nearing, G. (2019). Towards learning universal, regional, and local hydrological behaviors via machine learning applied to large-sample datasets. Hydrology and Earth System Sciences, 23(12), 5089-5110. https://hess.copernicus.org/articles/23/5089/2019/)

4. Lines 33 - 34: I am not sure if "flood forecasting" is really a synonym for "streamflow prediction / runoff forecasting". Furthermore, "runoff" usually describes water on an area that does not infiltrate or evaporate and, hence, discharges from that area; streamflow is the actual discharge that you measure in a river or channel. Thus, while these terms are very related, they do not mean the same thing.

**Answer:** Thank you for the comment. We have modified the sentence "Streamflow prediction and runoff modeling are modeling efforts where the water from the land or channel over time is being modeled"

5. Line 69: I found the wording here quite strange. You usually do not forecast measurements, but some phenomenon like precipitation or runoff.

**Answer:** We have updated to the "streamflow rate".

6. Line 120: "The WaterBench is not selected based on human activities, which is a reaction to the real situation in Iowa" --> What do you want to say here?

**Answer:** We have now removed this sentence. Some datasets (i.e., CAMELS) only contain the watersheds unaffected by human activity, which is simpler to forecast. However, flood impacts are also important in urban or suburban areas.

7. Lines 139 - 144: This sounds not very "hydrological". You should rather say that red dots are located at outlets of larger basins, which are divided into several smaller upstream sub- basins. And the outflow from each sub-basin is measured at the green gauging stations. See, e.g., http://proceedings.esri.com/library/userconf/proc01/professional/papers/pap1008/p1008.html for a good explanation of the terminology. For better visibility, it might make more sense to separate the larger basins from their sub-basins, e.g., by using thick and thin lines in Figure 1.

**Answer:** Thank you for your suggestions. We updated the sentences and figures.

8. Line 148: What are "statistics of the metadata"?

**Answer:** It is updated with "statistics of the data", including the basic statistical values such as min, max, average, and median value.

9. Line 149 - 150: "Metadata" usually refers to "data that provides information about data". You mean simply "static data". Please rephrase: For each basin, we provide static data (area, slope, travel time, ...) as well as time-series for streamflow, precipitation, and ET.

**Answer:** Thank you for the suggestion. We have rephrased them.

10.    Line 195: Your particular soil dataset has 12 soil-types. And I am pretty sure, that there are newer and much higher resolved maps available (see, e.g., https://www.cen.uni-hamburg.de/icdc/data/land/soilmap.html). Such data would fit better to your high-resolved precipitation data (even if you're only looking at basin averages).

**Answer:** Your information is very valuable. The soil map you mentioned is more detailed and in a higher resolution. We will verify and test this version of the soil dataset and include it in the future version.

11.    Line 206: Please re-phrase: For each station, streamflow data was aggregated to hourly values.

**Answer:** We have rephrased this sentence as you advised.

12.    Line 213: This sounds, once more, very "un-hydrological" and I strongly assume that you don't have to explain what "precipitation" is.

**Answer:** We have removed this sentence.

13.    Lines 206 - 207: Since there were... -> Please check your grammar... This sentence does not make sense.

**Answer:** We have rephased this sentence. "The original data contains a few missing values due to station system failures or internet outages."

14.    Lines 218 - 220: This sounds, again, overly complex... For "precipitation on the watershed", you usually say "basin-averaged precipitation".

**Answer:** We have revised this sentence as you advised. "In the dataset, we provide the hourly basin-averaged precipitation data for each station."

15.    Lines 222 - 223: Describing evapotranspiration as a major loss of precipitation sounds quite uncommon. Maybe say "precipitated water".

**Answer:** We have revised this sentence as you advised.

16.    Line 223: "no high-resolution ET dataset". What about ERA5 (Land)? GLEAM? MERRA? MSWX?

**Answer:** We have revised this to "no high-resoluation real-time ET dataset". Our final goal of the expected data-driven or deep learning model is to make real-time predictions. Thus, we did not include the reanalysis model output data that we cannot obtain in real-time. For example, ERA5-Land has a delay of 3 months, and GLEAM is only available till the end of 2021 (around 6 months delay).

17.     Line 224: What do you mean with "empirical dataset"?

**Answer:** For the evapotranspiration, the IFC has been using a simple climatology based on 12 years of North American Land Data Assimilation System. This approach captures the seasonal effects but fails to account for year-to-year and day-to-day variability.

18.     Line 252: Please rephrase: where Y_i is the observation at time i, Y^hat_i is the model result at time i, ...; and please add that sigma and mu refer to the forecasts while your Y refer to the observations! Furthermore, you usually use the same parameter "family" for the mean and standard deviation. So, either use Greek letters (as, e.g., in https://hess.copernicus.org/preprints/hess-2019-327/hess-2019-327.pdf), or stick to Latin letters (as in your Equation 1).

**Answer:** We have revised this sentence as you advised.

19.     Line 258: "Thus, a median value...": While this is certainly true, it also makes a lot of sense to analyse the distribution of your performance metrics across your 125 basins in order to get an idea where and why your model performs better/worse. Figure 4 is only a starting point for such an analysis.

**Answer:** We provided the standard deviation in the updated Table 4. We also included the CDF now. This is a benchmark paper to provide basic results, and we encourage the future researchers to beat our benchmark results with higher median NSE or KGE as we mentioned in the reply to your major comment #1.

20.     Line 260: Why are the 120hr ahead predictions the most important values?

**Answer:** It is easy to predict the streamflow for the next 1-6 hours since it would not change too much from the streamflow rate at hour 0. The model performance always decreases with the prediction time. Thus, the model efficiency at the 120hr ahead prediction is the most important.

21.     Line 297: I would consider 5-day-forecasts as "medium-range

**Answer:** We have updated it to medium-range in this manuscript as you advised.

22.    Figure 2: Presenting CDFs for static parameters is quite unusual. It would be more helpful if you show Histograms here.

**Answer:** We have updated the figures and now they are histograms in Figure 2.

23.    Table 3: You would rather say "Summary statistics for precipitation and streamflow". And, for better readability, please add the period during which these numbers were calculated.

**Answer:** We have revised the table as you advised. It is now "Summary statistics for precipitation and streamflow among 125 catchments from water year 2012 to 2018."

**Reviewer#3:**

This paper proposes a comprehensive reference dataset for streamflow prediction based on use in data-driven and machine learning studies, providing reference performance for more advanced deep learning architectures on datasets for comparative analysis. There is certainly a lack of unified reference data in earth science research, and these studies are very useful in being able to outline a general algorithm for defining them. A detailed description of the results is given in the study, but a more detailed description on the data, but also on the climatology of the study area should probably be reported to provide more details for researchers who intend to use WaterBench for deep learning research in hydrology.

**Specific Comments:**

1. Line 17: "To some extent, WaterBench makes up for the lack of unified benchmarks in earth science research. We highly encourage researchers to use the WaterBench for deep learning research in hydrology."

   **Suggestion**: I would replace this sentence with a summary of the results; it will be the good performance that will encourage.

**Answer:** We have revised this sentence as you advised.

2. Line 84: "*This dataset solves the difficulty of data acquisition and does not require domain knowledge from meteorology.*"
   **Question**: What the authors mean by this sentence: "*... does not require knowledge of the domain from meteorology?*"

**Answer:** Yes, and we have updated the descriptions as you suggested.

3. Line 89: "*Scientific advancement, intrinsically, is supposed to be cumulative, and in order to have better generalized deep learning-based flood forecasting models, scientists need to build on top of what their fellow researchers have done.*"
   **Comment**: Certainly, scientists need to build on what their fellow researchers have done, but they also have to improve on what has been done.

**Answer:** We have updated this sentence to "*scientists need to build **and improve** on what their fellow researchers have done*"

4. Line 90: "*We believe that this could only be done by using the same set of testing mechanisms ...*"

**Comment**: This sentence is true after being certain that everything has been done correctly.

**Answer:** Yes you are correct. We have rephrased the sentence.

5. Line 102: *"There remains a need for a dataset that is more convenient to use in deep learning research given that most of the deep learning researchers are not domain experts."*
   **Question**: What the authors mean by this sentence?

**Answer:** Our paper preprocesses some complex data, builds an optimization task that computer scientists can understand, and provides optimization goals (i.e. the metrics like NSE and KGE we want to maximize). As a result, data scientists who lack meteorological and hydrological knowledge can use our data for machine learning or AI modeling. Academic collaborations are common between domain experts and data scientists. Benchmark datasets support a new type of collaboration that involves meteorologists and hydrologists processing data before passing it on to computer scientists for artificial intelligence modeling and analysis.

6. Line 110: *"This study proposes a flood forecasting dataset that follows FAIR data principles ..."*
   **Suggestion**: I would write some references and details about FAIR.

**Answer:** We have rephrased this sentence. "Our dataset follows FAIR data principles, which means it is findable and accessible through DOI, and the data is richly described with references."

7. Line 115: *"... it could be used in other studies with similar goals, such as physically based modeling with physical equations."*
   **Suggestion**: I think "*with physical equations*" is not needed.

**Answer:** We have removed it as you advised.

8. Line 165: *"From the tables above, it is shown that our dataset is limited to a certain range of precipitation since it contains the catchments only in Iowa."*
   **Question**: What the authors mean by this sentence?

**Answer:** We have removed this sentence since we updated our title and descriptions that this is WaterBench-Iowa.

9. Line 167: *"Geologically, all the catchments are located in two HUC watersheds, the Upper Mississippi and Missouri, and the study results may not be applicable to other*

*regions in the U.S."*
**Question**: Do the authors think that this study is only about the analyzed area?

**Answer:** No. The trained models will be only applicable to the studied area, but the data-driven model and structures can be used in other regions around the world. We have rephrased this sentence.

10.  Line 168: *"WaterBench is also subject to a relatively high missing data rate for streamflow since the reliable hourly dataset is limited in USGS for some of the watersheds in Iowa."*
    **Question**: How have you thought about solving this limitation? How much data is missing?

**Answer:** This is mainly due to the freezing of rivers in winter, and we do not consider these data at present.

11.  Line 174: *"Since the total precipitation amount is the product of precipitation intensity and area, larger watersheds typically have higher streamflow rates."*
    **Comment**: I do not think this comment is needed, because river flow discharge depend on a number of conditions.

**Answer:** Yes. You are correct. We have rephrased this sentence.

12.  Line 179: *"2.2.2 Time of Concentration."*
    **Question**: Isn't the time of concentration calculated for each grid point of the considered domain? Is the velocity not estimated as a function of slope?

**Answer:**  The time of concentration is calculated based on the most distant point in the watershed to the outlet divide by an average velocity of 0.75m/s. This has been confirmed the 0.75m/s is very effective in Iowa basins and used in the following studies:

- Mantilla, R., Gupta, V. K. & Troutman, B. M., "Scaling of peak flows with constant flow velocity in random self-similar networks," *Nonlinear Processes in Geophysics*, vol. 18, no. 4, pp. 489–502, 2011.
- Mandapaka, P. V., Krajewski, W. F., Mantilla, R. & Gupta, V. K., "Dissecting the effect of rainfall variability on the statistical structure of peak flows," *Advances in Water Resources*, vol. 32, no. 10, pp. 1508–1525, 2009.
- Sloan, B. P., Mantilla, R., Fonley, M., & Basu, N. B., "Hydrologic impacts of subsurface drainage from the field to watershed scale," *Hydrological Processes*, vol. 31, no. 17, pp. 3017-3028, 2017.

- Fonley, M., Mantilla, R., Small, S. J., & Curtu, R., "On the propagation of diel signals in river networks using analytic solutions of flow equations," *Hydrology and Earth System Sciences*, vol. 20, no. 7, pp. 2899-2912, 2016.

13.  Line 188: *"A steep slope may cause a higher velocity and lower infiltration rate, which normally causes a larger streamflow rate at a precipitation event"*
**Suggestion**: Therefore, it would be very important to have the velocity data for the different considered grid points.

**Answer:** It is true and your suggestion is very helpful and we will include them in the future version of the benchmark.

14.  Line 206: *"Since there were a few missing values in the original data caused by station system breakdown or internet outages."*
**Question**: How is missing data handled?

**Answer:** In the dataset, we provided the original data with missing values. In our 120 hour prediction tasks, we removed the data with missing values. Users can fill missing values using interpolation.

15.  Line 221: *"2.2.7 Evapotranspiration (ET)."*
**Question**: Do the authors believe that this comprehensive reference dataset for streamflow prediction is intended for climate or meteorological studies? For climate studies the monthly ET parameter may be useful; for meteorological-hydro studies perhaps, it would be more functional to use soil moisture data (also useful for climate studies) and at a higher temporal resolution.

**Answer:** Our goal in creating this task is to implement real-time streamflow forecasts that take advantage of the efficient performance of machine learning. Thus, we did not include many meteorological features which cannot be measured real-time or near-real-time. Your suggestion is very helpful that the soil moisture data would be useful. We will include the  real-time hourly soil moisture data if there is one in our further studies.

16.  Line 145: *"Since general statistics such as mean squared error (MSE) and root mean squared error (RMSE) are not dimensionless..."*
**Suggestion**: However, it would be interesting to see some results.

**Answer**: Since we have 125 watersheds with the streamflow ranges from 10 cfs to 1000 cfs, we only can provide dimensionless statistics. However, we have provided the standard deviations and CDF plots in our revised manuscript.

17. Line 255: *"This means that there will be 120 different NSE and KGE values for different hours at each watershed."*
    **Question**: What the authors mean by this sentence?

**Answer**: Since we worked on the prediction up to 120 hours in the test year, we have the KGE and NSE model efficiency results from the 1-hr ahead predictions to 120-hr ahead predictions at the test year.

18. Line 256: *"It should be noted that since the watersheds here are not filtered, it is possible for some watersheds to be greatly affected by human activities, including mitigation, construction, irrigation, urban drainage, etc. activities in watersheds."*
    **Question**: Doesn't this sentence conflict with what is stated in line 120?

**Answer:** No. In line 120, we stated "The WaterBench is not selected based on human activities, which is a reaction to the real situation in Iowa." This means human activities in watersheds would be a challenge for prediction tasks (i.e., increasing soil erosion and more runoff with time). However, some benchmark dataset such as CAMELS **only** selected the rural areas without human activities.

19. Line 275: "Table 4. The median NSE and KGE among 125 watersheds at the prediction hour 1, 6, 12, 24, 48, 72, 96, and 120."
    **Question**: what do the second and third columns indicate?

**Answer:** Sorry for the improper formatting of the table. It is now fixed. The second and third columns are NSE values.

20. Line 287: "Table 5. The median NSE and KGE among 125 watersheds at the prediction hour 1, 6, 12, 24, 48, 72, 96, and 120."
    **Question**: what do the second and third columns indicate?

**Answer:** Sorry for the improper formatting of the table. It was an error of the table format and now fixed. The second and third columns are NSE values.

21. Line 297: *"The second limitation is the scale effect."*
    **Comment**: Unfortunately, it is not clear from the work what scales are being considered. Would the scale effects be the size of the drainage basin?

**Answer:** Yes. We rephased it as *"The second limitation is the scale effect, where the large basins have better model performance on the streamflow forecast and the small basins are hard to predict."*

22. Line 300: "The scale effect observed in our benchmark indicates the prediction on small watersheds is still a challenge."
    **Question**: Given the Classification of Watersheds by Size from the values in Table 2 it is evident the small to medium sized watersheds in the considered domain are a large number. I do not understand the statement made in line 300.

**Answer:** We have around half of the basins over 1000 km2 in Table 2, and it is observed that the larger watersheds are better overall performance from Figure 7.

23. Line 315: *"Although the data is limited to the Midwest, we believe that any studies on this dataset could provide insights for other streamflow forecasting and rainfall-runoff modeling studies at other watersheds."*
    **Comment**: This sentence written in this way is contrary to what is stated in line 168.

**Answer:** We have rephrased the sentence in Line 168. *"Geologically, all the catchments are located in two HUC watersheds, the Upper Mississippi and Missouri, and the study results may not be applicable to other regions in the U.S. However, the modeling algorithms and the neural network architectures normally apply to a broad spectrum of problems, and they would be useful in other regions.*

---

## Author Response (AR2)

**WaterBench: A Large-scale Benchmark Dataset for Data-Driven Streamflow Forecasting**

**Response to Reviewers' Comments**

ESSD-2022-52 | Data description paper
Submitted on 09 Feb 2022
Ibrahim Demir, Zhongrun Xiang, Bekir Demiray, and Muhammed Sit
Special Issue: Benchmark datasets and machine learning algorithms for Earth system science data (ESSD/GMD inter-journal SI)

**Topical Editor decision: Publish subject to minor revisions (review by editor)**

Comments to the author:

Dear authors,

thank you for submitting a detailed response to the 3 reviewers and a revised manuscript. Before accepting this paper for publication, I have a couple of minor points, which I ask you to improve:

1) the zenodo record which you refer to should contain a README file describing what exactly users will find in the csv and zip files. This description should include a link to the code repository and a link to itself (i.e. the zenodo doi) as well as to the paper in ESSD. And it should include a (brief) description of the file structure and format.

Answer: Thank you for the suggestion, and we have updated the Zenodo record with a readme file.

2) according to the ESSD publication rules, code must be provided in a non-alterable form, for example as another zenodo dataset with a doi. You can either upload a zip or tar archive of your code from git to the same zenodo record that contains the data, or you can create a new record. The link in the abstract should not point to a github repo. The github repo can be mentioned in the "Data and Code Availability" section like "The most recent code version can be found at https://github.com/uihilab/WaterBench.".

Answer: We have updated the abstract, data and code availability section as you suggested.

3) if you don't mind, I would suggest rephrasing the sentence on the domain knowledge again: "This dataset solves the difficulty of data acquisition and does not require domain knowledge of the domain of meteorology." to "While knowledge of the application domain is essential to find scientifically robust ways to prepare the input data and to interpret the results of machine learning models, such knowledge is not always accessible to deep learning experts. If there are well-defined benchmark datasets with a clear description of the machine learning task to solve and with well-defined and domain-science informed evaluation metrics, then it becomes possible for non-domain experts to solve such challenges and to introduce novel machine learning methods to the field."

- if you accept this suggestion, you may be able to shorten the text between lines 97 and 104 in your track changes manuscript.

Answer: We have rephrased and shortened the sentences as you suggested.

4) line 124: you can add https://www.nature.com/articles/sdata201618 as reference to FAIR data.

Answer: We have updated this citation.

5) the benchmark task itself should appear in the abstract - You can move the sentence "we define a sample benchmark task of predicting the hourly streamflow for the next five days for future comparative studies." from line 264 to the abstract and slightly rewrite the next sentence "The benchmark task defined here simulates rainfall..."

-- please rephrase that sentence, though, as "simulate ... forecasts ... real life" don't go well together.

Answer: We have updated the abstract, and rephrased the sentence as you suggested.

6) Acknowledgements should include the source of funding of this study and possibly a comment to thank the reviewers.

Answer: We have updated the acknowledgement as you suggested.